# Adaptive Formation Control for Underactuation Multi-USVs with Jointly Connected Switching Typologies*

1st Kunting Yu
*Navigation College*
*Dalian Maritime University*
Dalian, P. R. China
kuntingyu@163.com

2nd Yongming Li
*College of Science*
*Liaoning University of Technology*
Jinzhou, P. R. China
l_y_m_2004@163.com

3rd Kewen Li
*College of Science*
*Liaoning University of Technology*
Jinzhou, P. R. China
likewen2018@163.com

*Abstract*—This paper investigates the adaptive formation control problem for underactuation Multiply Unmanned Surface Vehicles (multi-USVs) connected by the jointly connected switching typologies. First, the distributed switching observer is devised for each USV to estimate the trajectory signal under the jointly connected switching typologies. Secondly, the underactuation controllers are designed based on the estimated trajectory signal under backstepping structure. In the form of the designed underactuated controller, we have proven through Lyapunov stability theory that each unmanned vessel is able to track the predetermined trajectory. In the end, we validate the effectiveness of the proposed method through numerical simulations.

*Index Terms*—switching typologies, underactuated input, multi-USVs, adaptive control.

## I. INTRODUCTION

In practical applications, communication networks often change over time for various reasons [1]. Many discussions have been carried out on the control problems of unmanned USVs under switched topologies [2]-[5]. [2] develops a data-driven neural predictor using real-time and historical data, leading to the design of an adaptive kinematic control law ensuring that the tracking error of each unmanned ship is consistently ultimately bounded. [3] designs a switching distributed extended state observer to estimate the unknown dynamics. Building upon this technique, [3] further devises distributed controllers for USVs to cooperatively track the target. The authors in [4], using orthonormal transformation technique and algebraic graph theory, equivalently formulates the formation control problem as analyzing the strictly dissipative problem of a Markov jump system with model-related time-varying delays. It proves the existence of dissipative martingale solutions. [5] treats Gaussian noise as a Wiener process, transforming the control problem of USV systems into a more specific stochastic problem. By utilizing the average dwell time method and Lyapunov stability theory, it proves that all signals of the USV system remain bounded in mean-square within a specified time. However, the aforementioned control

methods for unmanned USVs under switched topologies all necessitate each mode of the switched topology to have a spanning tree, significantly increasing the communication burden of the unmanned USVs under switched topologies.

In comparison to the aforementioned switched topologies modes, jointly connected digraphs only require the set of switched topology subsets to have a spanning tree in a certain interval of time. Therefore, jointly connected digraphs do not demand as many communication resources as the switching modes in [2]-[5]. This feature has made jointly connected digraphs a hot research area in the field of switching topologies [6]-[17]. [6] investigates the coordinated output regulation problem of linear multi-agent systems under undirected jointly connected topologies. Building upon this work, [7] and [8] further propose coordinated output regulation control methods for linear multi-agent systems under directed jointly connected topologies. The author in [9], by avoiding reliance on the leader's state, addresses the robust output regulation control problem for nonlinear models with unknown bounded disturbances under jointly connected topologies.

However, currently, there have been no reported results on jointly connected topologies for multi-USVs. This is due to the complex nonlinear dynamic characteristics of USVs, making it challenging to analyze their stability under jointly connected topologies. In order to address this gap, this paper proposes a formation control strategy for underactuated multi-USVs under jointly connected topologies. The innovations of this paper are as follows

- By designing a switching trajectory observer, this paper successfully transforms the formation control problem of multi-unmanned ships into a distributed trajectory tracking problem. Building upon the results of [8], we analyze the stability of trajectory observation errors under jointly connected topologies and provide a stability proof for the trajectory observation errors.
- Based on the trajectory estimation vector obtained from the trajectory observer, we design a distributed controller for underactuated unmanned ships, enabling each unmanned ship to track the trajectory based on the relative

This work was supported by the National Natural Science Foundation of China under Grants U22A2043. (Corresponding author: Yongming Li).

position in formation, thereby achieving the formation control effect.

The rest sections of this paper are organized as follows: Section II shows the problem formulation. In Section III, we give the primary stability results. Section IV depicts the simulation results. And Section V concludes the paper.

## II. PROBLEM FORMULATION

### A. Graph Theory

In this paper, the communication network of the $N$ USVs is modeled as a directed graph $\mathcal{P} = (\mathcal{G}, \mathcal{C})$, where the edge set is defined as $\mathcal{C} \subseteq \mathcal{G} \times \mathcal{G}$. If $j$-th USV can transmit information to $i$-th USV, symbol $a_{ij} \in \mathcal{V}$ is defined as 1. And if $j$-th USV can not transmit information to $i$-th USV, we define $a_{ij} = 0$. The Laplacian matrix $\mathcal{M}$ is designed as $\mathcal{M} = [m_{ij}] \in \mathbb{R}^{N \times N}$, where $m_{ij} = -a_{ij}$ and $m_{ii} = \sum_{j=1}^{N} a_{ij}$. Furthermore, symbol $b_i = 1$ is defined while $i$-th USV can receive the trajectory information. The matrices $\Delta$ and $\mathcal{B}$ are defined as $\Delta = \mathcal{M} + \mathcal{B}$ and $\mathcal{B} = \mathrm{diag}\{b_i\}$, respectively.

**Definition 1 (jointly connected graph)** [7]. There exist an infinite sequence of nonempty and non-overlapping time intervals $[t_l, t_{l+1})$ such that $t_{l+1} - t_l \leq T$ for a positive constant $T$, and a sequence of non-overlapping subintervals $[t_l^0, t_l^q), \cdots, [t_l^q, t_l^{q+1})$ with $t_l^0 = t_l$, $t_l^{q+1} = t_{l+1}$ and $T^* \leq t_l^{q+1} - t_l^q$ for a positive constant $T$, respectively. If there exist a time-varying switching signal $\sigma(t)$ and the union of the subgraphs $\bigcup_{p=0}^{q+1} \mathcal{P}_{\sigma(t_l^p, t_l^{p+1})}$ contains a spanning tree, then $\mathcal{P}_\sigma$ is the jointly connected.

**Assumption 1** [8]. In this paper, the directed graph $\mathcal{P}_\sigma$ is assumed to be jointly connected by setting $t_l^q$ as the switching time.

**Assumption 2** [10]. The trajectory signal $\beta_0(t) = [x_0, y_0]^T$ satisfies $\|\beta_0(t)\| \leq \beta_0^*$ and $\int_{t_0}^{t} \left\| \dot{\beta}_0(\tau) \right\| d\tau \leq \bar{\beta}_0$ such that $\beta_0^*$ and $\bar{\beta}_0$ are positive constants.

**Lemma 1** [8]. Under Assumption 1, if the system satisfies the following dynamic

$$\dot{\varsigma}(t) = -\varepsilon(\Delta \otimes I_n)\varsigma(t)$$

then we have

$$\|\varsigma(t)\| \leq -c \|\varsigma(t)\| e^{-\bar{c}(t-\tau)}, \forall t > \tau$$

where $\varepsilon$, $c$ and $\bar{c}$ are positive constants, $\varsigma \in \mathbb{R}^{Nn}$ is the system state vector.

**Lemma 2** (Bellman-Gronwall Lemma) [8]. When nonnegative piecewise continuous functions $g_1(t)$, $g_2(t)$, $g_3(t)$ and $g_4(t)$ satisfy

$$g_1(t) \leq g_2(t) + g_3(t) \int_{t_0}^{t} g_4(\tau)g_1(\tau)d\tau, t > t_0$$

then the following inequality holds

$$g_1(t) \leq g_2(t) + g_3(t) \int_{t_0}^{t} g_2(\tau)g_4(\tau)e^{\int_{\tau}^{t} g_4(s)g_3(\tau)d\tau} d\tau, t > t_0$$

### B. Problem Formulation

In this paper, the kinematics and dynamics of $i$-th USV is modeled as follows [18]

$$
\begin{aligned}
\dot{x}_i =& u_i \cos(\psi_i) - v_i \sin(\psi_i) \\
\dot{y}_i =& u_i \sin(\psi_i) - v_i \cos(\psi_i) \\
\dot{\psi}_i =& r_i \\
\dot{u}_i =& f_{i,1}(u_i) + \frac{m_{i,22}}{m_{i,11}} v_i r_i + \frac{1}{m_{i,11}}(\tau_{i,u} + \tau_{i,\omega u}) \\
\dot{v}_i =& -\frac{m_{11}}{m_{22}} ur + f_{i,2}(v_i) + \frac{1}{m_{i,22}}\tau_{i,\omega v} \\
\dot{r}_i =& f_{i,3}(r_i) + \frac{m_{11} - m_{22}}{m_{33}} u_i v_i + \frac{1}{m_{i,33}}(\tau_{i,r} + \tau_{i,\omega r})
\end{aligned}
\tag{1}
$$

where $(x_i, y_i)$ is denoted by the positions of each USV, and $\psi_i$ represents the yaw angle in relation to the earth-fixed frame. The velocities $u_i$, $v_i$, and $r_i$ correspond to the surge velocity, sway velocity, and yaw rate in the vehicle body-fixed frame, respectively. The disturbances $\tau_{i,\omega u}$, $\tau_{i,\omega v}$, and $\tau_{i,\omega r}$ are factors representing various environmental disturbances like wind, waves, and currents. These disturbances are assumed to be unknown, smooth, and bounded. Control inputs $\tau_{i,u}$ and $\tau_{i,r}$ are provided by the USV's propeller and rudder, respectively. They influence the motion of the vehicle. The dynamics of the system, encapsulated by $f_{i,1}(\cdot)$, $f_{i,2}(\cdot)$, and $f_{i,3}(\cdot)$, encompass a range of nonlinear effects including hydrodynamic damping forces, Coriolis centripetal forces, and other unmodeled hydrodynamic forces. The form of them can be found in [10].

*Control Objective*. The control objective of this paper is to design a distributed observer estimating the trajectory signal under jointly connected switching topologies. In addition, underactuated controllers are designed for the multi-USVs to follow the trajectory signal.

## III. MAIN RESULTS

In this section, we design the distributed observer and analysis the estimation performance. And then we design the underactuated controllers for multi-USVs to track the trajectory estimated signal.

### A. Distributed Switching Observer

In this paper, the distributed observer is designed as

$$
\begin{aligned}
\dot{\hat{\beta}}_{i,0} =& -\kappa_1 \hat{\beta}_{i,0} + \kappa_1 \hat{\beta}_{i,1} \\
\dot{\hat{\beta}}_{i,1} =& \hat{\beta}_{i,1} - \kappa_2 \sum_{j=1}^{N} a_{ij}^{\sigma(t)} (\hat{\beta}_{j,1} - \hat{\beta}_{i,1}) \\
& - \kappa_2 b_i^{\sigma(t)} (\beta_0 - \hat{\beta}_{i,1})
\end{aligned}
\tag{2}
$$

where $\kappa_1$ and $\kappa_2$ are designed positive constants, $\hat{\beta}_{i,0} = [\hat{x}_{i,0}, \hat{y}_{i,0}]^T$ and $\hat{\beta}_{i,1} = [\hat{x}_{i,1}, \hat{y}_{i,1}]^T$.

**Theorem 1**. The states of distributed switching observer can track the trajectory signal within a small neighborhood near the origin.

**proof**. First, we prove the estimation error as $\tilde{\beta}_{i,0} = \beta_0 - \hat{\beta}_{i,1}$ is bounded under jointly connected switching topologies. By taking the time derivative of the estimation error $\tilde{\beta}_{i,0} = \beta_0 - \hat{\beta}_{i,1}$, we have

$$\dot{\tilde{\beta}}_{i,0} = \dot{\beta}_0 - \hat{\beta}_{i,1} + \kappa_2 \sum_{j=1}^{N} a_{ij}^{\sigma(t)}(\hat{\beta}_{j,1} - \hat{\beta}_{i,1})$$
$$+ \kappa_{i,2} b_i^{\sigma(t)}(\beta_0 - \hat{\beta}_{i,1}) \tag{3}$$

By defining $\tilde{\beta}_0 = [\tilde{\beta}_{1,0}^T, \cdots, \tilde{\beta}_{N,0}^T]^T \in \mathbb{R}^{2N}$ and $\hat{\beta}_0 = [\hat{\beta}_{1,0}^T, \cdots, \hat{\beta}_{N,0}^T]^T \in \mathbb{R}^{2N}$, one yields

$$\dot{\tilde{\beta}}_0 = (\dot{\beta}_0 \otimes I_N) - \hat{\beta}_1 - \kappa_2(\Delta_{\sigma(t)} \otimes I_2)\tilde{\beta}_0 \tag{4}$$

On the basis of Lemma 1, we can obtain

$$\left\| \tilde{\beta}_0 \right\| \leq c \int_{t_0}^{t} \left\| \tilde{\beta}_0(\tau) \right\| e^{-\bar{c}(t-\tau)} d\tau$$
$$+ c \left\| \tilde{\beta}_0(t_0) \right\| e^{-\bar{c}(t-t_0)} \tag{5}$$
$$+ (1 + \sqrt{N}) c \bar{\beta}_0 \int_{t_0}^{t} e^{-\bar{c}(t-\tau)} d\tau$$

According to $\int_{t_0}^{t} e^{-\bar{c}(t-\tau)} d\tau = \frac{1}{\bar{c}} e^{-\bar{c}t}(e^{\bar{c}t} - 1) \leq \frac{1}{\bar{c}}$, it follows

$$\left\| \tilde{\beta}_0 \right\| \leq c \left\| \tilde{\beta}_0(t_0) \right\| e^{-\bar{c}(t-t_0)} + \frac{1}{\bar{c}}(1 + \sqrt{N})c\bar{\beta}_0$$
$$+ c \int_{t_0}^{t} \left\| \tilde{\beta}_0(\tau) \right\| e^{-\bar{c}(t-\tau)} d\tau \tag{6}$$

Furthermore, by employing Lemma 2, we can yield that

$$\left\| \tilde{\beta}_0 \right\| \leq c \left\| \tilde{\beta}_0(t_0) \right\| e^{-\bar{c}(t-t_0)} + \frac{1}{\bar{c}}(1 + \sqrt{N})c\bar{\beta}_0$$
$$+ c^2 \left\| \tilde{\beta}_0(t_0) \right\| \int_{t_0}^{t} e^{-2\bar{c}(t-\tau)} e^{\int_{\tau}^{t} e^{-\bar{c}(t-s)ds}} d\tau \tag{7}$$

Considering the method of integral comparison, the fact $\int_{t_0}^{t} e^{\int_{\tau}^{t} e^{-\bar{c}(t-s)ds}} d\tau \leq e^{\frac{1}{\bar{c}}} \int_{t_0}^{t} e^{-\bar{c}(\tau-t)} d\tau = \frac{1}{\bar{c}} e^{t}$ holds. Substituting this fact into (7), one has

$$\left\| \tilde{\beta}_0 \right\| \leq \hat{c} \left\| \tilde{\beta}_0(t_0) \right\| e^{-(\bar{c}-1)(t-t_0)} + \frac{1}{\bar{c}}(1 + \sqrt{N})c\bar{\beta}_0 \tag{8}$$

where $\hat{c} = \min\{c, \frac{c^2}{2\bar{c}^2}\}$. Thus, the estimation error $\tilde{\beta}_{i,0}$ is bounded for $t \in [0, \infty)$.

In what follows, we prove the state vector $\hat{\beta}_{i,0}$ can track $\hat{\beta}_{i,1}$. So that the state vectors $\hat{\beta}_{i,0}$ and $\hat{\beta}_{i,1}$ can estimate the trajectory signal $\beta_0$. Design the error as $\tilde{\beta}_{i,1} = \hat{\beta}_{i,0} - \hat{\beta}_{i,1}$ and Lyapunov function as $V_\beta = \sum_{i=1}^{N} \frac{1}{2}\tilde{\beta}_{i,1}^T \tilde{\beta}_{i,1}$, respectively. The time derivative of Lyapunov function $V_\beta$ can be calculated as

$$\dot{V}_\beta = \tilde{\beta}_1^T[-\kappa_1 \tilde{\beta}_1 - \kappa_2 \Delta_{\sigma(t)} \tilde{\beta}_0 + \tilde{\beta}_0 - \beta_0 \otimes I_2] \tag{9}$$

where $\tilde{\beta}_1 = [\tilde{\beta}_{1,1}^T, \cdots, \tilde{\beta}_{N,1}^T]^T \in \mathbb{R}^{2N}$ and $\hat{\beta}_1 = [\hat{\beta}_{1,1}^T, \cdots, \hat{\beta}_{N,1}^T]^T \in \mathbb{R}^{2N}$.

Under Assumption 1, one obtains

$$\dot{V}_\beta \leq - (\kappa_1 - 1)\tilde{\beta}_1^T \tilde{\beta}_1 + (\kappa_2 \left\| \Delta_{\sigma(t)} \right\| + 1)^2 \left\| \tilde{\beta}_0 \right\|^2$$
$$+ N\beta_0^{*2} \tag{10}$$

From (10), we can conclude that the estimation error $\tilde{\beta}_{i,1}$ is bounded for $t \in [0, \infty)$. Therefore, the states of distributed switching observer can track the trajectory signal within a small neighborhood near the origin. The proof of Theorem 1 is completed. ∎

### B. Distributed Underactuated Controller Design

In this subsection, the underactuated controllers for multi-USVs are designed to track the trajectory estimated signal $\hat{\beta}_{i,0}$. Define the position errors of each USV as $\tilde{x}_i = x_i - \hat{x}_{i,0} - l_i$ and $\tilde{y}_i = y_i - \hat{y}_{i,0} - \bar{l}_i$, the angle error as $z_{i,1} = \psi_i - \psi_{i,d}$, where $(l_i, \bar{l}_i)$ is the relative position of $i$-th USV in the formation, and $\psi_{i,d} = \arctan 2(\frac{\tilde{y}_i}{\tilde{x}_i})$. The concrete expression of $\arctan 2(\cdot)$ can be found in [18].

In what follows, we design the control input $\tau_{i,r}$ for underactuated USV model (1).

Step 1. The time derivative of angle error $z_{i,1} = \psi_i - \psi_{i,d}$ satisfies

$$\dot{z}_{i,1} = \dot{\psi}_i - \dot{\psi}_{i,d} = z_{i,2} + \alpha_{i,r} - \dot{\psi}_{i,d} \tag{11}$$

where $z_{i,2} = r_i - \alpha_{i,r}$, $\alpha_{i,r}$ is the virtual control signal for yaw rate $r_i$ under backstepping structure.

Design the Lyapunov function at this step as $V_{i,1} = \frac{1}{2} \ln \frac{z_{i,1}^2}{\nu_{i,\psi}^2 - z_{i,1}^2}$ with $\nu_{i,\psi} < \frac{\pi}{2}$. Its time derivative follows

$$\dot{V}_{i,1} = \frac{z_{i,1}}{\nu_{i,\psi}^2 - z_{i,1}^2}(z_{i,2} + \alpha_{i,r} - \dot{\psi}_{i,d}) \tag{12}$$

By applying Young's inequality, one yields

$$\frac{z_{i,1}}{\nu_{i,\psi}^2 - z_{i,1}^2}z_{i,2} \leq \frac{1}{2}\frac{z_{i,1}^2}{(\nu_{i,\psi}^2 - z_{i,1}^2)^2} + \frac{1}{2}z_{i,2}^2 \tag{13}$$

Substituting (13) into (12), we have

$$\dot{V}_{i,1} = \frac{z_{i,1}}{\nu_{i,\psi}^2 - z_{i,1}^2}(\alpha_{i,r} - \dot{\psi}_{i,d}) + \frac{1}{2}\frac{z_{i,1}^2}{(\nu_{i,\psi}^2 - z_{i,1}^2)^2}$$
$$+ \frac{1}{2}z_{i,2}^2 \tag{14}$$

Design the virtual controller as

$$\alpha_{i,r} = -c_{i,1}z_{i,1} + \dot{\psi}_{i,d} - \frac{1}{2}\frac{z_{i,1}}{\nu_{i,\psi}^2 - z_{i,1}^2} \tag{15}$$

where $c_{i,1}$ is a designed positive constant.

The final version of Lyapunov function $V_{i,1}$ can be expressed as

$$\dot{V}_{i,1} \leq \frac{-c_{i,1}z_{i,1}^2}{\nu_{i,\psi}^2 - z_{i,1}^2} + \frac{1}{2}z_{i,2}^2 \tag{16}$$

Step 2. The time derivative of error $z_{i,2} = r_i - \alpha_{i,r}$ can be expressed as

$$\dot{z}_{i,2} = \frac{m_{i,11} - m_{i,22}}{m_{i,33}} u_i v_i + f_{i,3}(r_i) + \frac{1}{m_{i,33}} \tau_{i,r} + \frac{1}{m_{i,33}} \tau_{i,\omega r} - \dot{\alpha}_{i,r} \tag{17}$$

By employing fuzzy logical systems (FLSs) to approximate the unknown dynamic $m_{i,11} - m_{i,22} u_i v_i + m_{i,33} f_{i,3}(r_i)$, (17) can be rewritten as

$$\dot{z}_{i,2} = \frac{1}{m_{i,33}} \theta_{i,1}^{*T} \varphi_{i,1}(u_i, v_i, r_i) + \frac{1}{m_{i,33}} \varepsilon_{i,1}(u_i, v_i, r_i) \\ + \frac{1}{m_{i,33}} \tau_{i,r} + \frac{1}{m_{i,33}} \tau_{i,\omega r} - \dot{\alpha}_{i,r} \tag{18}$$

where $\theta_{i,1}^*$ is the ideal vector of FLSs, $\varphi_{i,1}(u_i, v_i, r_i)$ is fuzzy basis function, and $\varepsilon_{i,1}(u_i, v_i, r_i)$ is the bounded approximation error.

Design the Lyapunov function at this step as $V_{i,2} = V_{i,1} + \frac{m_{i,33}}{2} z_{i,2}^2 + \frac{1}{2} \tilde{\Theta}_{i,1}^2$, where $\tilde{\Theta}_{i,1} = \Theta_{i,1} - \hat{\Theta}_{i,1}$, $\Theta_{i,1} = \theta_{i,1}^{*T} \theta_{i,1}^*$, and $\hat{\Theta}_{i,1}$ is the estimated variable of unknown product of FLSs vector $\Theta_{i,1} = \theta_{i,1}^{*T} \theta_{i,1}^*$. The time derivative of Lyapunov function $V_{i,2}$ can be calculated as

$$\dot{V}_{i,2} \leq \frac{-c_{i,1} z_{i,1}^2}{\nu_{i,\psi}^2 - z_{i,1}^2} + \frac{1}{2} z_{i,2}^2 + z_{i,2}[\theta_{i,1}^{*T} \varphi_{i,1}(u_i, v_i, r_i) \\ + \varepsilon_{i,1}(u_i, v_i, r_i) + \tau_{i,r} + \tau_{i,\omega r} - m_{i,33} \dot{\alpha}_{i,r}] \\ - \tilde{\Theta}_{i,1} \dot{\hat{\Theta}}_{i,1} \tag{19}$$

Invoking Young's inequality, the following inequalities hold

$$z_{i,2} \theta_{i,1}^{*T} \varphi_{i,1}(u_i, v_i, r_i) \leq \frac{\Theta_{i,1}^* z_{i,2}^2}{4\kappa \varphi_{i,1}^T(r_i) \varphi_{i,1}(r_i)} + \kappa \\ z_{i,2} \varepsilon_{i,1}(u_i, v_i, r_i) \leq \frac{1}{2} z_{i,2}^2 + \frac{1}{2} \varepsilon_{i,1}^{*2} \\ z_{i,2} \tau_{i,\omega r} \leq \frac{1}{2} z_{i,2}^2 + \frac{1}{2} \tau_{i,\omega r}^{*2} \tag{20}$$

Substituting (20) into (19), we can obtain

$$\dot{V}_{i,2} \leq \frac{-c_{i,1} z_{i,1}^2}{\nu_{i,\psi}^2 - z_{i,1}^2} + z_{i,2}[\frac{\Theta_{i,1}^* z_{i,2}}{4\kappa \varphi_{i,1}^T(r_i) \varphi_{i,1}(r_i)} \\ + \frac{3}{2} z_{i,2} + \tau_{i,r} - m_{i,33} \dot{\alpha}_{i,r}] + \kappa \\ + \frac{1}{2} \varepsilon_{i,1}^{*2} + \frac{1}{2} \tau_{i,\omega r}^{*2} - \tilde{\Theta}_{i,1} \dot{\hat{\Theta}}_{i,1} \tag{21}$$

In this step, design the controller $\tau_{i,r}$ and adaptive law $\dot{\hat{\Theta}}_{i,1}$ as

$$\tau_{i,r} = -(c_{i,2} + \frac{3}{2}) z_{i,2} - \frac{\hat{\Theta}_{i,1} z_{i,2}}{4\kappa \varphi_{i,1}^T(r_i) \varphi_{i,1}(r_i)} \\ + m_{i,33} \dot{\alpha}_{i,r} \\ \dot{\hat{\Theta}}_{i,1} = \frac{z_{i,2}^2}{4\kappa \varphi_{i,1}^T(r_i) \varphi_{i,1}(r_i)} - \gamma_{i,1} \hat{\Theta}_{i,1} \tag{22}$$

where $c_{i,2}$ and $\gamma_{i,1}$ are designed positive constants.

Substituting (22) into (21), the final version of Lyapunov function $V_{i,2}$ can be described as

$$\dot{V}_{i,2} \leq \frac{-c_{i,1} z_{i,1}^2}{\nu_{i,\psi}^2 - z_{i,1}^2} - c_{i,2} z_{i,2}^2 - \frac{\gamma_{i,1}}{2} \tilde{\Theta}_{i,1}^2 + \kappa \\ + \frac{1}{2} \varepsilon_{i,1}^{*2} + \frac{1}{2} \tau_{i,\omega r}^{*2} + \frac{\gamma_{i,1}}{2} \Theta_{i,1}^{*2} \tag{23}$$

In what follows, we design the control input $\tau_{i,u}$. To design the surge force, define the global position errors of $i$-th USV as $\tilde{z}_i = \sqrt{\tilde{x}_i^2 + \tilde{y}_i^2}$.

Step 1. Recalling the definitions of $\tilde{x}_i$, $\tilde{y}_i$, $\tilde{z}_i$ and $\psi_{i,d}$, we have $\tilde{x}_i = \tilde{z}_i \cos(\psi_{i,d})$ and $\tilde{y}_i = \tilde{z}_i \sin(\psi_{i,d})$. Therefore, the time derivative of $\tilde{z}_i$ satisfies

$$\dot{\tilde{z}}_i = u_i \cos(\tilde{\psi}_i) - v_i \sin(\tilde{\psi}_i) - \dot{x}_{i,0} \cos(\psi_{i,d}) \\ - \dot{y}_{i,0} \sin(\psi_{i,d}) \tag{24}$$

where $\tilde{\psi}_i = z_{i,1}$.

Construct the Lyapunov function at this step as $V_{i,3} = \frac{1}{2} \tilde{z}_i^2$. The time derivative of it can be described as

$$\dot{V}_{i,3} = \tilde{z}_i(u_i \cos(\tilde{\psi}_i) - v_i \sin(\tilde{\psi}_i) \\ - \dot{x}_{i,0} \cos(\psi_{i,d}) - \dot{y}_{i,0} \sin(\psi_{i,d})) \tag{25}$$

Define the backstepping error as $z_{i,3} = u_i - \alpha_{i,u}$, where $\alpha_{i,u}$ is the virtual control signal for surge velocity $u_i$. Therefore, (25) can be written as

$$\dot{V}_{i,3} = \tilde{z}_i[(z_{i,3} + \alpha_{i,u}) \cos(\tilde{\psi}_i) - v_i \sin(\tilde{\psi}_i) \\ - \dot{x}_{i,0} \cos(\psi_{i,d}) - \dot{y}_{i,0} \sin(\psi_{i,d})] \tag{26}$$

At this step, the virtual control $\alpha_{i,u}$ is designed as

$$\alpha_{i,u} = \frac{1}{\cos(\tilde{\psi}_i)}[-(c_{i,3} + \frac{1}{2}) \tilde{z}_i + v_i \sin(\tilde{\psi}_i) \\ + \dot{x}_{i,0} \cos(\psi_{i,d}) + \dot{y}_{i,0} \sin(\psi_{i,d})] \tag{27}$$

where $c_{i,3}$ is a designed positive constant.

Substituting (27) into (26), we get that

$$\dot{V}_{i,3} = -c_{i,3} \tilde{z}_i^2 + \frac{1}{2} z_{i,3}^2 \tag{28}$$

Note that $\cos(\tilde{\psi}_i)$ may be equal to 0, which will result in the singularity problem in (27). To avoid this problem, we construct the Lyapunov function as $V_{i,1} = \frac{1}{2} \ln \frac{z_{i,1}^2}{\nu_{i,\psi}^2 - z_{i,1}^2}$, which is called barrier Lyapunov function (BLF). If the BLF is bounded and the initial value $z_{i,1}(0)$ satisfies $|z_{i,1}(0)| \leq \nu_{i,\psi}$, it can yield that $-\frac{\pi}{2} < -\nu_{i,\psi} < z_{i,1}(t) < \nu_{i,\psi} < \frac{\pi}{2}$. From $\ln \frac{\nu_{i,\psi}^2}{\nu_{i,\psi}^2 - z_{i,1}^2} \leq \frac{z_{i,1}^2}{\nu_{i,\psi}^2 - z_{i,1}^2}$ and (23), choosing the positive constants as $\rho = \min\{c_{i,1}/2, c_{i,2}/2, \gamma_{i,1}\}$ and $\omega = \sum_{i=1}^N (\kappa + \frac{1}{2} \varepsilon_{i,1}^{*2} + \frac{1}{2} \tau_{i,\omega r}^{*2} + \frac{\gamma_{i,1}}{2} \Theta_{i,1}^{*2})$, we have $V_2 = \sum_{i=1}^N V_{i,2} \leq -\rho V_2 + \omega$. Based on this fact, we can conclude that $V_2$ is bounded and $z_{i,1}$ satisfies $-\frac{\pi}{2} < -\nu_{i,\psi} < z_{i,1}(t) < \nu_{i,\psi} < \frac{\pi}{2}$, $\forall t \in [0, \infty)$, respectively. Thus, there is no singularity problem in (27).

Step 2. Recalling the definition of backstepping error $\dot{z}_{i,3}$, the time derivative of it follows

$$
\begin{aligned}
\dot{z}_{i,3} =& \frac{m_{i,22}}{m_{i,11}} v_i r_i + f_{i,1}(u_i) + \frac{1}{m_{i,11}} \tau_{i,u} \\
&+ \frac{1}{m_{i,11}} \tau_{i,\omega u} - \dot{\alpha}_{i,u}
\end{aligned}
\tag{29}
$$

Invoking FLSs approximating the uncertain dynamic $m_{i,22} v_i r_i + m_{i,11} f_{i,1}(u_i)$, we have

$$
\begin{aligned}
\dot{z}_{i,3} =& \frac{1}{m_{i,11}} [\theta_{i,2}^{*T} \varphi_{i,2}(v_i, r_i, u_i) + \varepsilon_{i,2}(v_i, r_i, u_i)] \\
&+ \frac{1}{m_{i,11}} \tau_{i,u} + \frac{1}{m_{i,11}} \tau_{i,\omega u} - \dot{\alpha}_{i,u}
\end{aligned}
\tag{30}
$$

where $\theta_{i,2}^*$ is the ideal vector of FLSs, $\varphi_{i,2}(v_i, r_i, u_i)$ is fuzzy basis function, and $\varepsilon_{i,2}(v_i, r_i, u_i)$ is the bounded approximation error.

Consider the Lyapunov function at this step as $V_{i,4} = V_{i,3} + \frac{m_{i,11}}{2} z_{i,3}^2 + \frac{1}{2} \tilde{\Theta}_{i,2}^2$, the time derivative of it follows

$$
\begin{aligned}
\dot{V}_{i,4} \leq& -c_{i,3} \tilde{z}_i^2 + \frac{1}{2} z_{i,3}^2 + z_{i,3} [\theta_{i,2}^{*T} \varphi_{i,2}(v_i, r_i, u_i) \\
&+ \varepsilon_{i,2}(v_i, r_i, u_i) + \tau_{i,u} + \tau_{i,\omega u} - m_{i,11} \dot{\alpha}_{i,u}] \\
&- \tilde{\Theta}_{i,2} \dot{\hat{\Theta}}_{i,2}
\end{aligned}
\tag{31}
$$

where $\tilde{\Theta}_{i,2} = \Theta_{i,2} - \hat{\Theta}_{i,2}$, $\Theta_{i,2} = \theta_{i,2}^{*T} \theta_{i,2}^*$, and $\hat{\Theta}_{i,2}$ is the estimated variable of unknown product of FLSs vector $\Theta_{i,2} = \theta_{i,1}^{*T} \theta_{i,1}^*$

Applying Young's inequality, it follows

$$
\begin{aligned}
z_{i,3} \theta_{i,2}^{*T} \varphi_{i,2}(v_i, r_i, u_i) \leq& \frac{\Theta_{i,2}^* z_{i,3}^2}{4\kappa \varphi_{i,2}^T(u_i) \varphi_{i,2}(u_i)} + \kappa \\
z_{i,3} \varepsilon_{i,2}(u_i, v_i, r_i) \leq& \frac{1}{2} z_{i,3}^2 + \frac{1}{2} \varepsilon_{i,2}^{*2} \\
z_{i,3} \tau_{i,\omega u} \leq& \frac{1}{2} z_{i,3}^2 + \frac{1}{2} \tau_{i,\omega u}^{*2}
\end{aligned}
\tag{32}
$$

Substituting (32) into (31), we obtain

$$
\begin{aligned}
\dot{V}_{i,4} =& -c_{i,3} \tilde{z}_i^2 + z_{i,3} [\frac{\Theta_{i,2}^* z_{i,3}}{4\kappa \varphi_{i,2}^T(u_i) \varphi_{i,2}(u_i)} + \frac{3}{2} z_{i,3}^2 \\
&+ \tau_{i,u} - m_{i,11} \dot{\alpha}_{i,u}] + \kappa + \frac{1}{2} \varepsilon_{i,2}^{*2} + \frac{1}{2} \tau_{i,\omega u}^{*2} \\
&- \tilde{\Theta}_{i,2} \dot{\hat{\Theta}}_{i,2}
\end{aligned}
\tag{33}
$$

In this step, design the controller $\tau_{i,u}$ and adaptive law $\dot{\hat{\Theta}}_{i,2}$ as

$$
\begin{aligned}
\tau_{i,u} =& -(c_{i,4} + \frac{3}{2}) z_{i,3} - \frac{\hat{\Theta}_{i,2} z_{i,3}}{4\kappa \varphi_{i,2}^T(u_i) \varphi_{i,2}(u_i)} \\
&+ m_{i,11} \dot{\alpha}_{i,u}
\end{aligned}
\tag{34}
$$

$$
\dot{\hat{\Theta}}_{i,2} = \frac{\hat{\Theta}_{i,2} z_{i,3}^2}{4\kappa \varphi_{i,2}^T(u_i) \varphi_{i,2}(u_i)} - \gamma_{i,2} \hat{\Theta}_{i,2}
$$

where $c_{i,4}$ and $\gamma_{i,2}$ are designed positive constants.

Substituting (34) into (33), the final version of Lyapunov function $V_{i,4}$ can be described as

$$
\begin{aligned}
\dot{V}_{i,4} =& -c_{i,3} \tilde{z}_i^2 - c_{i,4} z_{i,3}^2 + \kappa + \frac{1}{2} \varepsilon_{i,2}^{*2} + \frac{1}{2} \tau_{i,\omega u}^{*2} \\
&- \frac{\gamma_{i,2}}{2} \tilde{\Theta}_{i,2}^2 + \frac{\gamma_{i,2}}{2} \Theta_{i,2}^{*2}
\end{aligned}
\tag{35}
$$

**Theorem 2**. The angle tracking error $z_{i,1}$ and position tracking error $\tilde{z}_i$ can converge to a small neighborhood near the origin.

**Proof**. Choosing the global Lyapunov function $V = \sum_{i=1}^N (V_{i,2} + V_{i,4})$, we can obtain

$$
\begin{aligned}
\dot{V} =& \sum_{i=1}^N [-\frac{c_{i,1} z_{i,1}^2}{\nu_{i,\psi}^2 - z_{i,1}^2} - c_{i,2} z_{i,2}^2 - \frac{\gamma_{i,1}}{2} \tilde{\Theta}_{i,1}^2 + \kappa \\
&+ \frac{1}{2} \varepsilon_{i,1}^{*2} + \frac{1}{2} \tau_{i,\omega r}^{*2} + \frac{\gamma_{i,1}}{2} \Theta_{i,1}^{*2} - c_{i,3} \tilde{z}_i^2 \\
&- c_{i,4} z_{i,3}^2 + \kappa + \frac{1}{2} \varepsilon_{i,2}^{*2} + \frac{1}{2} \tau_{i,\omega u}^{*2} - \frac{\gamma_{i,2}}{2} \tilde{\Theta}_{i,2}^2 \\
&+ \frac{\gamma_{i,2}}{2} \Theta_{i,2}^{*2}] \\
\leq& -CV + D
\end{aligned}
\tag{36}
$$

where $C = \min\{\frac{c_{i,1}}{2}, \frac{c_{i,2}}{2}, \frac{c_{i,3}}{2}, \frac{c_{i,4}}{2}, \gamma_{i,1}, \gamma_{i,2}\}$ and $D = \sum_{i=1}^N \{2\kappa + \frac{1}{2} \varepsilon_{i,1}^{*2} + \frac{1}{2} \tau_{i,\omega r}^{*2} + \frac{\gamma_{i,1}}{2} \Theta_{i,1}^{*2} + \frac{1}{2} \tau_{i,\omega u}^{*2} + \frac{\gamma_{i,2}}{2} \Theta_{i,2}^{*2}\}$.

## IV. SIMULATION RESULTS

In this section, three USVs are considered with the kinematics and dynamics model (1) to prove the effectiveness of the proposed method. Their parameters are referred [19], and the switching typologies communication networks are depicted in Fig. 1, respectively. The switching interval is designed to be 0.3s. From Fig. 1, we can observe that the considered switching typologies do not need that each switching mode $\mathcal{P}_{\sigma(t)}$ involves a spanning tree, which is different from the existing switching results on USVs [1]-[5].

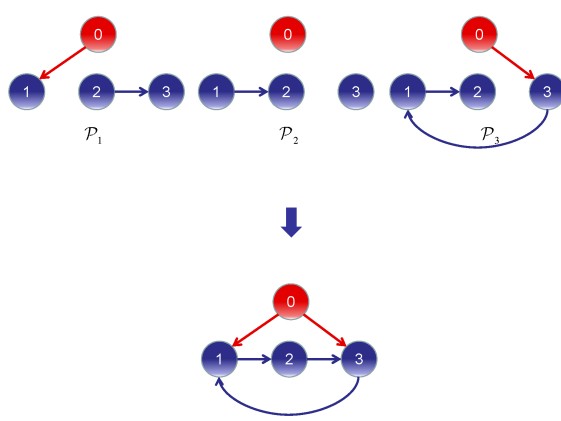

Fig. 1. The considered switching typologies communication networks.

The considered trajectory is designed as $\beta_0 = [60\cos(\pi t/60), 60\sin(\pi t/60)]^T$. The designed parameters of distributed observer are chosen as $\kappa_1 = \kappa_2 = 200$. And the

other designed parameters of (15), (22), (27) and (34) are set as $c_{i,1} = 15$, $c_{i,2} = 20$, $c_{i,3} = 40$, $c_{i,3} = 30$ and $\nu_{i,\psi} = \frac{\pi}{2}$.

The simulation results are shown in Figs. 2-3. Fig. 2 shows the path trajectories of each USV. It can be seen that each USV can follow the desired path with their own formation position. The formation mission is achieved. Fig. 3 displays the trajectories of angle tracking error $z_{i,1}$. From Fig. 3, we can conclude that each angle tracking error $z_{i,1}$ are constrained within the designed boundary $\nu_{i,\psi}$, which demonstrates that the virtual controller (27) avoids the singularity problem.

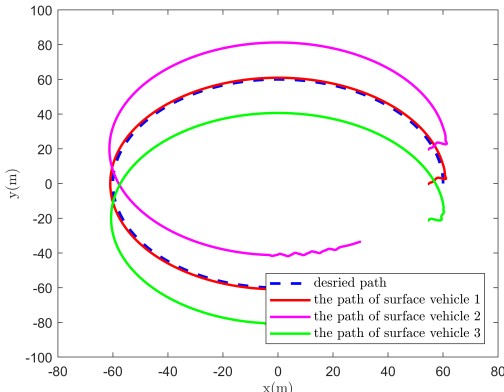

Fig. 2. The path trajectories of each USV.

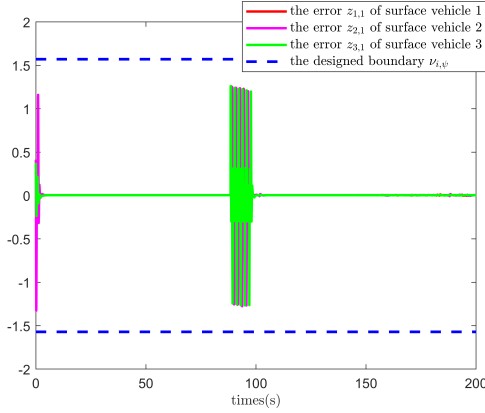

Fig. 3. The trajectories of angle tracking error $z_{i,1}$.

## V. Conclusion

This paper proposes a formation control method for Underactuation multi-USVs with jointly connected switching typologies. To provide the trajectory signal for each USV, distributed switching observer is designed. And by employing the signal of distributed switching observer, the underactuated inputs are devised under backstepping structure. In theory, we prove the estimation error and formation error can converge to a small neighborhood near the origin. Simulation results also demonstrate the effectiveness of the proposed method. In practice, finite-time control methods can ensure that unmanned vessels achieve stability within a finite time frame. Therefore, in the next phase of our research, we will extend this method to the field of finite-time control.

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
