# OpenReview forum: "Adaptive Formation Control for Underactuation multi-USVs with Jointly Connected Switching Typologies"
_IEEE.org/ICIST/2024/Conference — IEEE ICIST 2024 Conference Submission_

### Official Review · Reviewer_nobq · 2024-08-24
**This article is very interesting, but there are still some shortcomings.**

**Rating:** 7
**Confidence:** 4

**Review:**

The conclusion chapter is also too brief. I believe the author can provide relevant opinions on the usefulness and limitations of this method.

---

### Official Review · Reviewer_qMT9 · 2024-08-27
**The topic under consideration is interesting. This paper can be accepted after minor modifications.**

**Rating:** 8
**Confidence:** 3

**Review:**

This paper considers the adaptive formation control problem for underactuation Multiply Unmanned Surface Vehicles connected by the jointly connected switching typologies. The topic under consideration is interesting. Detailed comments and suggestions are listed as follows.
1.	The English writing of the paper needs to be further polished, and some typos should be fixed, such as “the distributed observer is design as…”, “Taking time derivative of it follows”, and “On the basis of the fact …, choosing the positive constants as…, is follows…”.
2.	In Page 5, the derivation of (31) cannot be easily followed. Please give a detailed explanation.
3.	The references should be added for provid ing a more comprehensive summary of the current research status.

---

### Official Review · Reviewer_xoG3 · 2024-08-30
**Accept**

**Rating:** 10
**Confidence:** 5

**Review:**

1.	If lemmas 1 and 2 are not first proposed by the author, the corresponding references need to be added.
2.	In the introduction, the author mentions some comparisons with existing studies (e.g. [2]-[5]) but does not summarize in detail the main contributions of this study and the specific differences with existing studies. It is suggested to add a paragraph to the introduction outlining the innovations of the paper.
3.	It is suggested that the author further provide the analysis of the learning effect of the network under different topology switching modes, especially the influence of the joint-connected topology on the stability and control accuracy of the USV system.

---

### Decision · Program_Chairs · 2024-09-06

Accept (Oral)